# Prevention of Pathological Fracture of the Proximal Femur: A Systematic Review of Surgical and Percutaneous Image-Guided Techniques Used in Interventional Oncology

**DOI:** 10.3390/medicina55120755

**Published:** 2019-11-22

**Authors:** Laëtitia Rodrigues, François H. Cornelis, Nicolas Reina, Sylvie Chevret

**Affiliations:** 1INSERM UMR 1153, Team ECSTRRA, Department of Biostatistics and Medical Information, AP-HP Saint Louis Hospital/Paris Diderot Université, 75010 Paris, France; sylvie.chevret@aphp.fr; 2Department of Radiology, Tenon Hospital, Sorbonne Université, 4 rue de la Chine, 75020 Paris, France; francois.cornelis@aphp.fr; 3Department of Orthopedic Surgery, Pierre-Paul-Riquet Hospital, CHU de Toulouse, 31300 Toulouse, France; nicolas.reina@chu-toulouse.fr

**Keywords:** proximal femur, bone metastasis, prevention, osteosynthesis

## Abstract

*Background and objectives:* Patients suffering from bone metastasis are at high risk for pathological fractures and especially hip fractures. Osteolytic metastases can induce a high morbidity rate (i.e., pain, facture risk, mobility impairment), and operation on them can be difficult in this frail population having a reduced life expectancy. Several medical devices have been investigated for the prevention of these pathological hip fractures. *Materials and Methods:* To investigate these solutions, a literature review and a meta-analysis of primary studies was performed. Data sources included electronic databases (PubMed, CENTRAL and ClinicalTrials.gov) from 1990 until 1 January 2019. Titles, abstracts and full-text articles were reviewed in order to select only studies evaluating the performance of the studied solution to prevent osteoporotic and/or pathological hip fracture. The main outcomes were the occurrence of hip fracture, pain evaluation (VAS score) and adverse events occurrence (including severe adverse events and deaths). All randomised controlled trials (RCTs) and cohort studies were considered. A Bayesian cumulative meta-analysis was undertaken on the primary studies conducted in patients with bone metastasis. *Results:* A total of 12 primary studies were identified, all were cohort studies without a control group, and one compared two devices, and were thereafter considered separately. In those 12 samples, 255 patients were included, mean age 61.7 years. After implantation, the cumulative risk of fracture was 5.5% (95% confidence interval, 3.0% to 8.6%), and adverse event occurrence was 17.4% (95%CI, 12.6 to 22.8%), with a median follow-up of 10 months. The posterior probability of a fracture rate below 5% was 40.3%. *Conclusions:* The literature about medical devices evaluation for preventing hip fractures in metastatic patients is poor and mostly based on studies with a limited level of evidence. However, this systematic review shows promising results in terms of efficacy and tolerance of these devices in patients with bone metastases. This treatment strategy requires further investigations.

## 1. Introduction

Bone is the third location of metastases after lungs and liver. Bone metastasis, further to a cancer (in particular breast, kidney, lung, prostate or thyroid [1,2,3]), are frequently located in the proximal part of the femur [4] and specifically in the femoral neck [5], highly exposed to fractures. Osteolytic metastases can induce a high morbidity rate due to abnormal bone resorption (i.e., pain, facture risk, mobility impairment) [6], and operation on them can be difficult in this frail population having a reduced life expectancy [7].

In these cancer patients, to prevent the occurrence of a pathological fracture of the proximal femur, surgical stabilization by femoroplasty or osteosynthesis is the first-line treatment, but only when patients can tolerate it, as it is associated with non-negligible surgical morbidity and mortality. The alternative for these patients is the use of less invasive techniques (compared to surgery) which use a percutaneous approach with small incisions, have an effective analgesic effect and allow for a potential reduction of disability after the procedure and of the risk of bleeding or infectious complications. Bed rest and hospital stay are also reduced, and complementary therapies can still be used such as radiation, thermal ablation and chemotherapy [8]. Prophylactic fixation is recommended in the case of lytic metastases of the proximal femur presenting a Mirels’ score above 8 to quantify the risk of sustaining a pathological fracture [9]. According to the literature, cementoplasty alone is effective for pain palliation but contraindicated in the proximal femur because of inadequate bone consolidation during weight bearing, thus not reducing the risk of occurrence of secondary fracture [10]. Therefore, minimally invasive solutions have been studied to be combined with bone cement for the treatment and/or prevention of these pathological hip fractures, such as cement-augmented metallic materials and percutaneous image-guided screw-mediated osteosynthesis [11].

In the last few years, several attempts have been made to develop the prevention of femoral neck fracture, in order to achieve immediate reinforcement in non-surgical patients. Studied techniques include the use of screws, nails as well as needles, and an innovative polymer implant was also developed, all combined with bone cement. However, these studied solutions have to provide evidence of performance without rising new risks linked to the invasive treatment. In this context, a literature review was conducted to evaluate the results of the current available solutions (on the market or experimental) for the prevention of pathological hip fracture due to bone metastasis.

## 2. Materials and Methods

### 2.1. Search Strategy and Selection Criteria

To investigate these solutions, a literature review was performed. Data sources included three electronic databases (PubMed, CENTRAL and ClinicalTrials.gov) from 1990 until 1 January 2019, using the following keywords: hip fracture prevention; impending pathological hip fracture. Reference lists of selected articles and known studies were also searched for additional references. In addition, specific researches were done about products already in the market for prevention (i.e., Piccolo proximal femur nail). All abstracts were exported, and duplicates were removed, then assessed to identify papers for full review. Titles, abstracts and full-text articles were reviewed in order to select all studies evaluating the performance of the studied solution to prevent pathological hip fracture. The main outcomes being the number of hip fractures, pain evaluation (VAS score), adverse events occurrence (including severe adverse events and deaths). Epidemiological, medico-economic and adverse-events studies were excluded. After duplicates removal, the articles were sorted according to the studied treatment, to keep only those on surgical medical devices. All randomised controlled trials (RCTs) and cohort studies were considered, as well as every other relevant study, depending on the available data (i.e., innovative-treatment starting clinical trials). Studies with no clinical measure in vivo (i.e., biomechanical studies) were excluded. Only records with full texts available and written in English were included.

### 2.2. Data Extraction and Quality Assessment

Records were sorted according to the study design (with or without a control group). For each included study, the following data were extracted into Excel worksheets using a pre-specified analysis grid, recording general information (name of the article, authors), study descriptors (control group, medical device name, type of study, type of centre), patient characteristics (age, sex, disease) and outcomes measures (number of fracture, mean VAS score, adverse events type and number, number of deaths, mean follow-up). The variables were described with mean and percentage. Further analyses were done on implanted groups to provide an estimation of the outcomes (number of fractures, adverse events and severe adverse events). Several variables (percentage of male, mean age, type of medical device being a nail, addition of cement, number of patients) were assessed for a possible association with these events’ occurrence using multiple logistic regression and exact Fisher test, as appropriate. The results were described with odds ratio with 95% confidence interval. Finally, the Newcastle–Ottawa assessment scale (NOS) was used for assessing the quality of these studies [12]. This gold-standard scale is based on three domains: selection of study groups, comparability of groups and ascertainment of exposure/outcome, with a maximum of 9 stars for case-control or cohorts’ studies. Note that, in the absence of a control group, the scale ends at 7 stars.

### 2.3. Systematic Review and Statistical Analysis

A systematic review was performed according to the guidelines of Preferred Reporting Items for Systematic Reviews and Meta-Analyses (PRISMA) (REF Moher, D PLOS 2009) of the primary studies in patients with bone metastasis that reported the estimated effect of surgical medical devices on clinical outcomes (hip fractures and adverse events). To provide estimates of fracture or adverse event (AE) prevalence in the treatment arms, a Bayesian cumulative meta-analysis was then performed [13]. Through Bayesian statistics, we estimated cumulatively the trial results according to the time of publication, after ordering the studies according to their publication time. Non-informative Beta priors (i.e., uniform priors) were used to represent the large uncertainty with regards to the outcomes before any published trial data. Then, the posterior distributions computed after each study were used as the priors for the next trial, and so on. The posterior mean probability of each event (fracture and AE) with its 95% credible interval and the posterior probability that the risk was less than 5% and 10% were reported. All point estimates are presented with 95% credibility intervals and were computed using Markov Chain Monte Carlo (MCMC) simulation. The risk of publication bias was assessed by funnel plots of effect estimates against sample size [14]. All computations were performed on R version 3.2.2 (https://www.R-project.org/), using the R2jags (https://cran.r-project.org/web/packages/R2jags/index.html).

## 3. Results

The preliminary search resulted in a total of 56 records identified through database searching, including references and specific researches. After removal of the duplicates, 51 articles where screened on the basis of titles and abstracts, and 39 articles were excluded. The reasons of exclusion were a wrong study population (*n* = 32), the absence of clinical measures in vivo (*n* = 3) or of performance evaluation (*n* = 3) and the lack of full text (*n* = 1) (Figure 1). A total of 12 articles were finally included in this review, representing 255 patients, of whom 234 were implanted with a surgical medical device, and 21 had no device implanted (control group, with only bone cement). The studies included no level I study, one level II, prospective comparative cohort, no level III studies and eleven level IV case series.

Regarding the main outcomes, all articles reported the number of fractures as well as adverse events (severe or not) and mean follow-up. The number of deaths was missing in one article, and the mean VAS score was recorded in only five articles. Information extracted from these articles is summarized in Table 1, including the rating score according to NOS.

Twelve studies presenting solutions for impending and pathological hip fracture due to bone metastasis were included in this review. Among these studies, 11 included 215 patients who received a surgical medical device for prevention without a control group (level IV studies), and one cohort study included 19 implanted patients with needles and cement compared with 21 control patients receiving cement alone [23] (level II study). Therefore, we only considered the implanted group of this study (*n* = 19) in the analysis. In these patients, several surgical medical devices were studied: screws (*n* = 3), polymer implant (*n* = 1), nails (*n* = 6) and needles (*n* = 2). Among the 234 implanted patients, the mean age was 61.7 years, and 47.9% of them were women. The mean follow-up was 10 months (range 7 months–1.5 years), with a hip fracture occurrence of 2.7% (ranging from 0 to 11.1) and death occurrence of 51.2%.

The fracture occurrence was recorded when using a nail [17,18,20] (9.4%, 2.4% and 11.1%, respectively) or with the polymer implant (10%) [15]. Similarly, the mean adverse event occurrence was 13.1%, rather homogenous among all studies; the highest rates of severe adverse events reported (i.e., premature death, fracture at insertion, deep infection) were from the same three nail studies (9.4%, 9.5% and 22.2%, respectively), as well as from a study using needles to reinforce the hip [23]. VAS score was evaluated in 5 out of 12 studies, with a mean of 1.7 at a mean time of 5 months (range 0–12). The results of univariable logistic regression models (Table 2) did not show any relationships between the studied variables (percentage of males, mean age, type of medical device being a nail, addition of cement, number of patients) and the outcomes (fracture, adverse event, severe adverse event).

A meta-analysis was then performed on these 12 primary samples. As reported in Figure 2, there was no evidence of any publication bias, with funnel plots including all but one of the estimates. Figure 3 displays the estimated proportion of hip fracture and of adverse events as data accumulated over time. On the basis of the total available data, with a median follow-up of 10 months, the risk of fracture was estimated at 5.5% (95% credible interval, 3.0% to 8.6%), and that of adverse event was 17.4% (95%CI, 12.6 to 22.8%). The posterior probability of a fracture rate below 5% was 40.3%.

## 4. Discussion

This literature review investigated a total of 12 articles, representing studies evaluating the performance of the studied solution in preventing pathological hip fracture due to bone metastasis. A total of 255 patients were included, of whom 234 patients had medical device implantations, and 21 patients had no implantations. Performance was assessed through the occurrence of hip fracture and adverse events. However, quality of life and pain evaluation were poorly examined.

The implanted devices are developed for cancer patients presenting with bone metastasis to overcome the risk of complications due to impeding fracture in patients with poor clinical status and life expectancy but also to improve the quality of life of these patients. Percutaneous cementoplasty is done for painful bone metastasis, when systemic treatment (i.e., chemotherapy or radiotherapy) is not enough to control pain and when the risk of fracture is low, but its efficacy in long bones is discussed because of a lack of stability and the occurrence of complications if revision is needed [6,23]. Prevention of bone metastasis using devices is gaining interest. Indeed, among the 12 studied articles, only 4 were published in 13 years (1991–2004), whereas 8 were published in the last 6 years (2012–2017). Preliminary results from various studies evaluating the benefits of augmented osteoplasty of the proximal femur demonstrate their feasibility, safety and efficiency in preventing pathological fracture (low rate of secondary pathological fracture) [10]. This technique is known to give effective pain palliation, in combination or not with other techniques, but the rate of fractures reported in the literature is high. All devices are minimally invasive, often introduced percutaneously under fluoroscopy guidance. They require local or general anaesthesia, and the hospital stay is short. Of the 12 selected studies in this meta-analysis, including 255 treated patients, the cumulative risk of fracture after a median follow-up of 10 months was 5.5% (95% CI, 3.0% to 8.6%), and the occurrence of adverse event was 17.4% (95%CI, 12.6 to 22.8%), relatively stable over time and cohort size; moreover, no factor of heterogeneity in patient subsets (age and sex) or devices (nails or cement) that could explain the occurrence of these events was found. The posterior probability of a fracture rate below 5% was 40.3%, making minimally invasive prevention a promising solution to reduce pain and the risk of pathological hip fracture in cancer patients and therefore enhance their quality of life. Besides, immunotherapy treatment has disrupted the management of these patients in the last few years, prolonging life expectancy and thus allowing surgical consolidation when needed. A recent review reported a life expectancy without cancer progression three time superior than the mean survival in all patients, compared to chemotherapy or targeted therapy [24]. Prophylactic fixation is recommended in cases of lytic metastases of the proximal femur presenting a Mirels’ score superior or equal to 8 to quantify the risk of sustaining a pathological fracture. Interventional oncology may be indicated depending on patient’s performance status and life expectancy, but it can also induce higher morbidity [23]. In cancer patients, these minimally invasive procedures appear therefore as a promising alternative, but the limited current evidence reported in the literature justifies a further evaluation of these treatments in a larger population.

This review presents several limitations. First, the level of evidence is limited, as the selected studies are mainly cases series and non-controlled, non-randomized trials which present a high risk of bias with imprecise results. Second, the choice of focusing only on surgical prevention of bone metastasis led to a limited number of selected articles: the preliminary search resulted in a total of 56 records identified through database searching, and only 12 studies were finally included. This review did not consider other interventions that might also reduce the risk of oncologic fractures such as cementoplasty alone, but this solution is already widely studied in the literature and used in current practice.

## 5. Conclusions

The literature about medical devices evaluation for preventing hip fractures in cancer patients is poor and mostly based on studies with a limited level of evidence. However, this systematic review shows that attention is being focused on the use of these devices in the presence of bone metastasis, with promising results in terms of efficacy and tolerance, which requires further investigations.

## Figures and Tables

**Figure 1 medicina-55-00755-f001:**
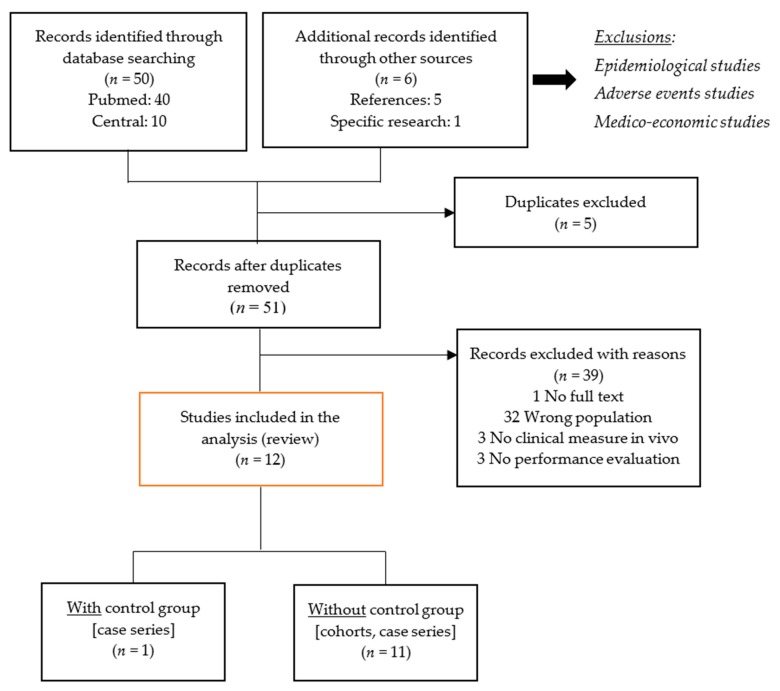
Methodology flowchart.

**Figure 2 medicina-55-00755-f002:**
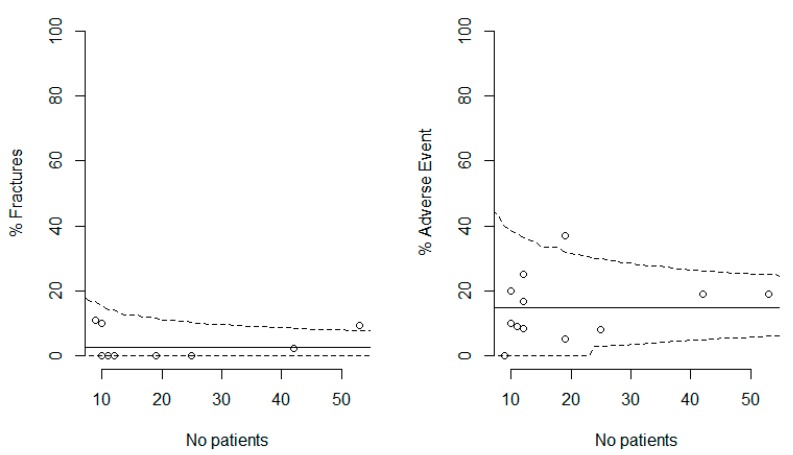
Funnel plots for the two study end points, namely, prevalence of hip fractures (left plot) and prevalence of adverse events (right plot) reported in the 12 studies. The outer dashed lines indicate the triangular region within which 95% of studies are expected to lie in the absence of both biases and heterogeneity. Note that, given that three of the eight studies with no reported fracture shared the same sample size (*n* = 12), only five distinct estimates at 0% fracture appear on the left plot.

**Figure 3 medicina-55-00755-f003:**
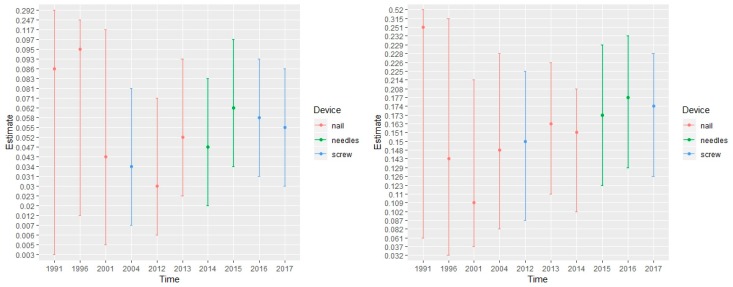
Posterior estimates with 95% credibility intervals of the prevalence of hip fracture (left plot) and of adverse event (right plot) over time of publication of the primary studies; the underlying types of medical device assessed in these studies are indicated by the colour of each estimated point.

**Table 1 medicina-55-00755-t001:** Description of the individual studies. AEs: adverse events, SAEs: severe adverse events, NK: not known.

Ref	Study	NOS stars (out of 9)	Medical Device	Cement	Number of Patients	% Man	Mean Age	Mean Follow-up	% Fracture	% AEs	% SAEs
[15]	Cornelis 2017	6	Implant	Yes	10	67	62	0.65	10	10	NK
[16]	Shemesh 2014	6	Nail	No	19	38.1	62	0.8	0	5.3	NK
[17]	Alvi 2013	6	Nail	No	53	42	60	0.9	9.4	18.9	9.4
[18]	Moholkar 2004	6	Nail	No	42	28.6	66	1.1	2.4	19	9.5
[19]	Edwards 2001	6	Nail	No	25	44	68	1.5	0	8	8
[20]	Voggenreiter 1996	6	Nail	Yes	9	22	59	0.7	11.1	0	22.2
[21]	Weikert 1991	6	Nail	No	10	NK	67	0.7	0	20	NK
[11]	Cazzato 2017	5	Screw	Yes	11	54.5	63.7	0.17	0	9.1	0
[22]	Deschamps 2012	6	Screw	Yes	12	25	55	0.64	0	16.7	0
[6]	Kelekis 2016	6	Needles	Yes	12	83	NK	1.35	0	25	0
[23]	Tian 2015	8	Needles	Yes	19	47.4	58.5	0.75	0	17.5	12.5
[8]	Mavrovi 2017	6	Screw	Yes	12	75	56	1.05	0	8.3	0

**Table 2 medicina-55-00755-t002:** Multivariate logistic regression models for predicting the outcomes in patients with bone metastasis. Nb: number of.

Endpoints	OR * (95% CI)	*p* Value
**Fracture**		
% male	0.96 (0.89;1.04)	0.32
Mean Age	1.02 (0.76;1.37)	0.91
Nail	1.67 (0.15;18.9)	0.68
Cement	0.60 (0.05;6.79)	0.68
Nb patients	1.08 (0.97;1.21)	0.16
**Adverse Event**		
% male	0.98 (0.92;1.05)	0.59
Mean Age	1.12 (0.83;1.52)	0.46
Nail	5.33 (0.38;75.8)	0.22
Cement	0.19 (0.01;2.66)	0.22
Nb patients	1.17 (0.90;1.55)	0.24
**Severe Adverse Event**		
% male	0.93 (0.85;1.02)	0.13
Mean Age	1.08 (0.81;1.45)	0.59
Nail	3.75 (0.33;42.5)	0.29
Cement	0.27 (0.02;3.02)	0.29
Nb patients	1.25 (0.93;1.69)	0.14

* OR (95% CI): odds ratio (95% confidence interval).

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
