# Peer review of "Prevention of Pathological Fracture of the Proximal Femur: A Systematic Review of Surgical and Percutaneous Image-Guided Techniques Used in Interventional Oncology"

_1010-660X, 2019, doi:10.3390/medicina55120755_

Round 1

Reviewer 1 Report

This paper systematically reviewed the correlation between hip fracture and medical device implantation for bone metastasis intervention. From heterogeneous datasets over the last thirty years, the author carefully selects study cohorts by only including the randomized controlled trials with clinical measures in vivo.

Major:
Details of measurements, study cohorts, missing data were documented in the text. However, the discrepancies between Table1 and Figure 2 are not explained. In Table1, there are eight studies reported 0% of fracture, however in Figure2A, there are only five studies plotted at the 0 marks, and the total studies plotted is only 10. In the meantime, the Figure2B included all 12 studies. It would be helpful if the author can provide details of the exclusion criteria specifically for figure 2A and 2B.

Minor:
1. The prediction outcomes in patients with bone metastasis are modeled using a multivariant logistic regression model in Table2 and Figure3. It’s hard to understand the OR of SAE in the Nail group has an upper bond of infinite. Is this a missing data or computing error?

2. The writing is cohesively reasoned with occasional break sentences (e.g, line 144-145). For the discussion in relation of Figure 2 and 3, the author could comment on the stable AE rate fracture estimation over time or cohort size.

Author Response

Review Report Form

Open Review

English language and style

( ) Extensive editing of English language and style required
( ) Moderate English changes required
(x) English language and style are fine/minor spell check required
( ) I don't feel qualified to judge about the English language and style

Is the work a significant contribution to the field?

Is the work well organized and comprehensively described?

Is the work scientifically sound and not misleading?

Are there appropriate and adequate references to related and previous work?

Is the English used correct and readable?

Comments and Suggestions for Authors

This paper systematically reviewed the correlation between hip fracture and medical device implantation for bone metastasis intervention. From heterogeneous datasets over the last thirty years, the author carefully selects study cohorts by only including the randomized controlled trials with clinical measures in vivo.

Major:
Details of measurements, study cohorts, missing data were documented in the text. However, the discrepancies between Table1 and Figure 2 are not explained. In Table1, there are eight studies reported 0% of fracture, however in Figure2A, there are only five studies plotted at the 0 marks, and the total studies plotted is only 10. In the meantime, the Figure2B included all 12 studies. It would be helpful if the author can provide details of the exclusion criteria specifically for figure 2A and 2B.

Answer: We agree with the Reviewer that there are 8 studies with no reported fracture.  However, 3 of those 8 studies shared the same sample size, that is, n=12; this explains why only 5 estimates of 0% appeared in the funnel plot (Figure 2A). This has been more clearly stated in the revised manuscript. Moreover, we agree that studies selected for those plots should be more clearly reported. Notably, left plot previously included results from the control group of the Tian (2005) study; it has been deleted from the revised figure. Both revised funnel plots refer to estimates from the 12 studies. This has been detailed in the revised manuscript.

Minor:
1. The prediction outcomes in patients with bone metastasis are modeled using a multivariant logistic regression model in Table2 and Figure3. It’s hard to understand the OR of SAE in the Nail group has an upper bond of infinite. Is this a missing data or computing error?

Answer: Actually, only univariable logistic regression models were fitted, owing to the small number of events (either fractures, adverse events or severe adverse events) that preclude the fit of such multivariable models. We thank the reviewer for his comment regarding the OR of SAE in the Nail group. Actually, we apologize because the reporting estimate was based on 3 studies with nail, all of which had severe adverse events (explaining the previously reported confidence interval); when data has been corrected, estimates were more reasonable, that is, OR at 3.75 (95%CI, 0.33 to 42.5). All results from Table 2 have been carefully checked, and typos corrected.

2. The writing is cohesively reasoned with occasional break sentences (e.g, line 144-145). For the discussion in relation of Figure 2 and 3, the author could comment on the stable AE rate fracture estimation over time or cohort size.

Answer: This has been added in the revised Discussion section of the manuscript.

Submission Date

17 October 2019

Date of this review

23 Oct 2019 02:39:33

Reviewer 2 Report

manuscript title: Prevention of Pathological Fracture of The Proximal Femur: A Systematic Review of Surgical and Percutaneous Image-Guided Techniques Used in Interventional Oncology

Expert oppinion:

The paper is well written and sound. The message is clear and updated.

Certainly it does not represent a breakthrough at any rate but it is a useful contribution to clear the ideas of the non specialized readers. 

Author Response

Review Report Form

Open Review

English language and style

( ) Extensive editing of English language and style required
( ) Moderate English changes required
(x) English language and style are fine/minor spell check required
( ) I don't feel qualified to judge about the English language and style

Is the work a significant contribution to the field?

Is the work well organized and comprehensively described?

Is the work scientifically sound and not misleading?

Are there appropriate and adequate references to related and previous work?

Is the English used correct and readable?

Comments and Suggestions for Authors

manuscript title: Prevention of Pathological Fracture of The Proximal Femur: A Systematic Review of Surgical and Percutaneous Image-Guided Techniques Used in Interventional Oncology

Expert opinion:

The paper is well written and sound. The message is clear and updated.

Certainly, it does not represent a breakthrough at any rate but it is a useful contribution to clear the ideas of the non-specialized readers. 

Submission Date

17 October 2019

Date of this review

28 Oct 2019 18:51:12

Round 2

Reviewer 1 Report

The authors successfully addressed all of the reviewers' comments. Overall, the manuscript meets the publication requirement.